# Laser Sensing and Vision Sensing Smart Blind Cane: A Review

**DOI:** 10.3390/s23020869

**Published:** 2023-01-12

**Authors:** Chunming Mai, Dongliang Xie, Lina Zeng, Zaijin Li, Zhibo Li, Zhongliang Qiao, Yi Qu, Guojun Liu, Lin Li

**Affiliations:** 1College of Physics and Eletronic Engineering, Hainan Normal University, Haikou 571158, China; 2Key Laboratory of Laser Technology and Optoelectronic Functional Materials of Hainan Province, Hainan Normal University, Haikou 571158, China

**Keywords:** smart blind cane, laser sensing, vision sensing, fusion integrated sensing, SLAM, target identification

## Abstract

Laser sensing and vision sensing smart canes can improve the convenience of travel for the visually impaired, but for the present, most of the system functions of laser sensing and vision sensing smart canes are still defective. Guide equipment and smart blind canes are introduced and classified first, and the smart blind canes based on vision sensing, laser sensing and laser vision sensing are investigated, respectively, and the research status of laser vision sensing smart blind canes is sorted out. The advantages and disadvantages of various laser vision sensing smart blind canes are summarized, especially the research development of laser vision fusion as the core of new smart canes. The future development prospects of laser vision sensing smart blind cane are overviewed, to boost the development of laser vision sensing smart blind cane, to provide safe and efficient travel guarantee for the visually impaired.

## 1. Introduction

Statistics in the first World Vision Report [1] released by the World Health Organization (WHO) in 2019 show that more than 2.2 billion people are visually impaired or blind worldwide. In 2015, there were an estimated 253 million people with visual impairment worldwide. Of these, 36 million were blind and 217 million suffer from moderate to severe visual impairment [2]. According to data released by the China Association of the Blind in 2019, there are 17.31 million visually impaired people in China, accounting for 1.2 percent of China’s total population and 20 percent of the world’s visually impaired. According to the Basic Situation Report of Visually Impaired Internet Users in China released by the China Information Accessibility Products Alliance in 2018, 30% of visually impaired in China basically choose to stay indoors and 15% choose to travel short distances due to visual limitations.

At present, the visually impaired usually use the blind track, guide dog, cane and other guide equipment to help them travel. However, when the visually impaired go to the unfamiliar and complex environments alone, the guide dog and cane can not provide the visually impaired with the visual information a normal person can get from the same situation, which will cause the visually impaired unable to walk. In the daily travel of the visually impaired, the common cane is widely used as an auxiliary device for them due to its convenience to carry and use, but because of its limitations in many aspects of use, the visually impaired face great travel challenges. Because the visually impaired can only feel the rough situation of the environment by tapping the front or around with the ordinary cane, the road information they can get while walking is very limited and the measuring range of the ordinary cane is limited, which can only cover the scope of about one meter around the visually impaired.

From 1960s to now, positioning and navigation technology have been introduced into the field of guide for the blind [3] and have experienced three stages of development: sensor technology, computer image processing technology, computer vision and speech recognition technology. Any guide equipment should at least be able to detect the distance of obstacles, the steps and pits, and the obstacles of low height [4]. Moreover, it should be able to provide clear and reliable environmental data within a few seconds, have consistent working performance during the day and at night, be used both indoors and outdoors, and detect static and dynamic objects [3].

Different guide assistive technologies constitute different guide systems mounted on different guide equipment. For example, the guide system is equipped with the cane [5], navigation belt [6], guide glasses [7], guide cap [8] (navigation helmet [9]), hand-worn guide instrument [10], head-mounted guide instrument [11], guide robot [12] (guide suitcase [13]), electronic guide dog [14], guide drone [15], guide mobile device (smartphone [16], tablet computer [17]) and other carriers to form different guide equipment. Through these guide devices, the visually impaired can obtain the direction and distance information of pedestrians and obstacles on the road as well as the road condition information that needs attention.

Among many guide equipment for the blind, the intelligent cane, also called smart blind cane, or smart cane, is highly valued by the visually impaired due to its stable structure and high practicability. The smart cane can be seen as composed of the main cane body and the guide system mounted on the cane (including input module, environmental data acquisition module, main control module, feedback module or driving module), as shown in Figure 1. Its input module enables the visually impaired to input the object they want to go to or identify into the system in the form of instructions through voice microphone, keyboard, button, mobile phone APP, WeChat mini program and other input devices, so that the smart cane can obtain working instructions. Its environmental data acquisition module (camera, Lidar, ultrasound, infrared, millimeter wave radar, RFID, GPS, etc.) is the core module of the smart cane, which plays the most important role in the working of the smart cane. It is used to collect environmental data around the visually impaired. The main control module (Arduino, STM32 micro-controller, Raspberry Pi, Jetson Nano) and other small central processing units are used to analyze and process the ambient data collected and transmit the instructions to the feedback module or drive module. Its feedback module (steering gear, motor, buzzer, voice module) is used to give feedback environmental information and navigation instructions to the visually impaired in the form of sound, touch, voice, vibration, etc. Some of the smart canes with active navigation ability will drive the moving device at the bottom (omnidirectional wheel, robot car) through the driving module, and guide the visually impaired to the destination.

On the basis of the traditional ordinary cane or smart cane, micro-controller and a variety of sensors have to be added in order to provide feedback road information and navigation information for the visually impaired. At present, the commercial smart canes can be divided into two categories: (1) Electronic Travel Aid (ETAs), such as ultrasonic, millimeter wave, infrared, IMU, vision, laser smart canes, etc., which can collect the environmental information of the visually impaired; (2) Electronic Orientation Aid (EOAs), such as GPS and GIS smart cane, which can determine the exact location of the visually impaired. Since the initial development of smart guide technology, ultrasonic sensor [18], infrared sensor [19], camera [20], Lidar [21], laser ranging scanner [22], millimeter-wave radar [23], color sensor [24], electronic compass [25], (IMU) Inertial Measurement Unit [26], (RFID)Radio Frequency Identification [27], GPS (Global Positioning System) [28] and GIS (Geographic Information System) [29], VR technology [30], virtual cane [31] and other sensors or schemes are flooded into the research and development of smart canes. Since these schemes use a single or two sensors, the function of this kind of smart cane is relatively simple, and it can only obtain certain information about obstacles in the environment around the visually impaired, such as direction, distance, shape, size, color, position and others, but no comprehensive environmental data around. If the smart cane is intended to complete the navigation and target recognition functions, multiple sensors are needed to collect data in all aspects of the environment. Therefore, in recent years, the sensing methods of the smart canes tend to develop in the direction of smart solutions, sensor integration and multi-sensor fusion. Therefore, it is a worthy research direction in the future to enable the intelligent blind cane to acquire the sensing capability that integrates laser sensing and visual sensing technologies, to acquire such functions as active navigation and guidance, efficient identification and detection of targets.

## 2. Application Status of Laser Vision Sensing Smart Cane

Because laser vision can help to get the information data of the surrounding environment efficiently, laser vision technology has played a huge role in the field of blind guide in recent years. This chapter introduces in detail the related programs globally for the research and development of laser sensing smart cane and vision sensing smart cane in recent years and makes a comparative analysis of the performance of various kinds of smart canes.

### 2.1. Laser Sensing Smart Cane

Laser sensing technology (including Lidar and laser rangefinder) is applied to guide equipment [32,33,34], which helps improve the ranging accuracy and distance of guide equipment. The laser sensor takes the laser as the signal source, which emits a laser beam from the laser, hitting the obstacle in front of the visually impaired, which is accepted by the receiver after the reflection, this way, the distance between the laser sensor and the obstacle can be measured. The three-dimensional structure information of obstacles can be accurately drawn according to the reflected energy and spectrum amplitude, frequency and phase of obstacles.

With the development of laser technology, the laser sensing smart blind cane, as a laser sensing guide equipment, has gradually improved its laser sensing performance. D Bolgiano et al. (1967) [35] mounted laser sensor on the blind cane for the first time, and JM Benjamin Jr et al. (around 1970) [4,36] developed C1-C5 laser smart blind cane at the same time. These early laser canes used laser sensors to detect obstacles in the path of the visually impaired and to respond with audio warnings and vibration signals when the person approached the obstacle. However, these early laser sensing smart canes were limited by the technology at that time, resulting in limited detection accuracy and range, and did not incorporate other sensors to make up for the defects of laser sensors, unable to map the surrounding environment. Therefore, in the follow-up development of laser smart canes, researchers have made unremitting efforts to improve the performance of laser smart canes in three aspects: improving the detection accuracy and detection range, integrating laser smart canes with other sensors, and developing laser SLAM smart canes with navigation mapping ability.

#### 2.1.1. Improvement in the Detection Accuracy and Range of the Laser Smart Canes

In order to improve the detection accuracy and range of the laser smart blind canes, R Farcy et al. (2006) [22] proposed a laser blind cane named Teletact II, in which the Teletact handheld laser telemetry instrument was mounted on the blind cane. The laser smart blind cane can detect obstacles within the range of 10 cm to 10 m in front of it, with 1% accuracy and a measurement rate of 40 times per second. However, the laser smart blind cane is easily disturbed by natural light, and its accuracy will be reduced in the outdoor environment even with good weather conditions. A Dernayka et al. (2021) [37] proposed smart blind cane named Tom Pouce III, as shown in Figure 2. The cane was equipped with a laser detector and an infrared sensor, which worked independently of each other. The laser detector can detect the obstacle 12 m away, but the detection is limited to 6 m if only the infrared sensor is used. The Tom Pouce III is less susceptible to interference from natural light than the Teletact II laser blind cane and can still detect obstacles in strong sunlight conditions. However, when the visually impaired wants to avoid the obstacles 4–50 cm away from the front at the normal walking speed, they need to move the blind cane slowly. This is because the detection response rate of the blind cane is insufficient, resulting in reduced movement rate of the visually impaired. To solve the problem of night navigation, F Felix et al. (2022) [38] proposed a navigation system for the visually impaired based on Lidar detection and ranging. The smart blind cane system detects obstacles along the way for the visually impaired and feeds information back to the visually impaired through buzzer and vibration motor. The system is also equipped with light-emitting diodes that light up when someone or something approaches the visually impaired, avoiding nighttime accidents in dark environments. When the obstacle is 1.2 m away from the smart cane, the system will vibrate to remind the visually impaired of the obstacle in front of him. When the obstacle is 0.6 m away from the smart cane, the system will send an alarm signal. In the development of laser smart blind canes, researchers try to develop laser sensing virtual blinds to replace the solid laser sensing smart blind canes. T Palleja et al. (2010) [31] proposed to develop a biological electronic smart blind cane based on the principle of tentacles. They loaded URG-04LX Lidar and triaxial accelerometer on the forearm tactile belt. The environment information is converted into tactile information and fed back to the visually impaired. It can detect obstacles within 6 m of the visually impaired and can achieve close-range navigation function. This kind of laser sensing virtual blind cane is similar to a hand-worn laser guide using laser scanning sensor. The laser beam emitted by the virtual blind cane replaces the body of the smart blind cane, which is equivalent to a longer detection distance and a wider detection range. However, this kind of virtual blind cane is critically dependent on the performance of laser sensor or other sensors, once the sensor error occurs or battery power is exhausted, the visually impaired will be in a helpless situation, because of the loss of any assistance, unable to avoid obstacles and unable to go to the destination. Although the detection accuracy and detection range of this kind of laser ranging smart blind cane have been improved, its single function is not suitable for use in complex road conditions, and it even needs to be paired with other sensors to work together to make up for the defects.

#### 2.1.2. Smart Blind Cane Composed of Laser and Other Sensors

As relying only on a single laser sensor can only measure the distance to obstacles and lack other necessary navigation functions, while the laser sensor coupled with other sensors mounted on the smart cane can is necessary to obtain more diversified environmental data to optimize the performance of the laser smart cane.

JA Hesch et al. (2010) [39] proposed a smart blind cane composed of a 3-axis gyroscope (IMU) and a 2D laser scanner. By integrating time changing IMU measurements to track the pose of the visually impaired, the smart cane system uses two-level estimators. In the first level, the line features detected by the laser scanner are used to detect the pose and direction of the smart cane; in the second level, the corner features detected by the laser scanner are used to estimate the indoor position of the visually impaired. The system needs a known indoor environment map to calculate the intersection of the laser scanning plane and the environmental structure plane. The measuring distance of the smart cane is 0.02~4 m, and the detection angle coverage range is 240 degrees. Y Tsuboi et al. (2019) [40] proposed a smart blind cane composed of a laser ranging sensor, an ultrasonic sensor and a voice coil motor. Laser sensors are used to detect platform steps, ultrasonic sensors are used to detect obstacles, and the voice coil motors are used to remind the visually impaired that there are obstacles or steps in front of him. The integration of the laser ranging sensor and the ultrasonic sensor can effectively increase the measuring distance range of the smart cane, and the ultrasonic sensor can make up for the defect that the laser ranging sensor cannot measure the obstacles in a short distance (e.g., within 1 m). Its measuring distance is 0.02~5.5 m, the detection range angle coverage is 15 degrees.

#### 2.1.3. Laser SLAM Smart Cane

SLAM is short for Simultaneous Localization and Mapping, which means simultaneous localization and map construction. SLAM technology based on Lidar has the advantages of accurately measuring the angle and distance of obstacle points and generating environment map which is easy to navigate. It has become an indispensable technology in the navigation and positioning scheme of mobile platforms at present.

In order to enable the laser smart cane to acquire navigation mapping capability, Slade P et al. (2021) [21] proposed an active smart cane based on laser SLAM technology. Two-dimensional Lidar, monocular camera, IMU and GPS module were installed on the cane, as shown in Figure 3. The smart cane system can output the position, speed, direction and surrounding obstacles of the visually impaired in real time. An omnidirectional wheel driven by a motor is installed at the bottom of the smart blind cane. The omnidirectional wheel can keep contact with the indoor and outdoor ground at all times to achieve left-right movement. A path planning algorithm is used to calculate a track to guide the visually impaired to bypass obstacles and reach the destination. The smart cane can effectively improve the walking speed of users by about 20%. However, its main control module is Raspberry Pi 4B, which uses the monocular camera Raspberry Pi official camera. Therefore, the target recognition ability and recognition efficiency of the smart cane are not high.

To sum up, various kinds of laser sensing smart blind canes have both advantages and disadvantages, as shown in Table 1. The laser SLAM smart blind cane has better performance than the laser ranging smart blind cane in terms of navigation mapping ability, and the laser sensor with other sensors mounted on the smart blind cane can obtain more diverse environmental data. In the future, the development of laser smart blind cane will definitely tend to have a wider detection range, longer detection distance and higher detection accuracy. The future development would be towards laser fusion SLAM smart blind cane, which integrates multiple sensors with the ability of navigation mapping.

### 2.2. Visual Sensing Smart Cane

The smart blind cane based on vision has been introduced to help the blind navigate and find his way. The visual sensors such as monocular camera, stereo camera, RGB-D camera and 3D-TOF camera can help to obtain the image information of the surrounding environment of the visually impaired by using optical elements and imaging devices to achieve image processing, target detection and tracking, and visual SLAM functions. The research and development of visual sensing smart canes can roughly be classified into four types, namely, visual SLAM smart cane, visual recognition smart cane, visual fusion sensing smart cane, and VR/MR smart cane. This section investigates the four types of visual intelligence canes, analyzes and summarizes their advantages and disadvantages.

#### 2.2.1. Visual SLAM Smart Cane

With the development of robotics, machine learning, deep learning and other artificial intelligence fields, visual SLAM technology has also achieved rapid development. The principle of vision SLAM is that a mobile platform equipped with vision or laser sensors was used to build a model of the surrounding environment to estimate its own position and motion state in the process of movement without any environmental prior conditions. The framework of visual SLAM technology consists of five parts: visual sensor information reading, front-end visual odometer, back-end (nonlinear) optimization, loop detection and map building. The application of visual SLAM in the smart blind cane can provide path planning and navigation obstacle avoidance capabilities. At present, smart blind canes with visual SLAM ability mostly use A* algorithm to plan the path and cooperate with IMU to improve their navigation performance.

In order to solve the problem that the system can hardly complete navigation tasks only by relying on visual SLAM, PF Alcantarilla et al. (2012) [41] proposed the concept of dense scene flow applied by visual SLAM. Through dense scene flow and adding the moving object detection module, the system can effectively obtain more accurate location and mapping results in a highly dynamic environment. However, the hardware part of the system needs to be worn on the visually impaired, and an external communication cable needs to be connected to the computer, so it is difficult to walk with the visually impaired. Moreover, if the system operates without moving target detection, the error between the motion trajectory estimated by the camera and the actual trajectory will be large. The navigation ability of visual SLAM can also be effectively improved by using visual inertia system. L Jin et al. (2020) [42] proposed a navigation aid device based on the iPhone 7 smartphone Visual Inertial odometer (VIO), which uses the iPhone 7 as a sensing and computing platform. The iPhone 7′s rear camera and LSM6DSR-iNEMO Inertial measurement unit serve as imaging and motion sensors to enable visual inertial fusion SLAM, which allows the system to plan the shortest path possible to a destination based on the location of the visually impaired and a pre-stored 2D plan. Similarly, Google Tango device is used to replace iPhone 7 to achieve visual SLAM function. Q Chen et al. (2017) [43] proposed a smart blind cane system named CCNY, which is based on Google Tango device, using visual inertial odometer (VIO) for positioning and path planning, using voice or scroll button to specify the desired destination, use A* algorithm to plan the path, and then using the sense of touch to feedback the navigation instructions to the visually impaired, to help the visually impaired navigate to the destination in an unfamiliar or complex indoor environment.

In order to make visual SLAM technology more suitable for the pathfinding system of navigation for the visually impaired in indoor environment, H Zhang et al. (2017) [44] developed a real-time pathfinding navigation system of indoor visual SLAM for the visually impaired based on SR4000 3D-TOF camera. A new two-step pose image SLAM method is proposed, which is better than the plane-based SLAM method. The two-step pose image SLAM method can reduce the 6-DOF pose error of the equipment with less computing time. The system uses A* algorithm for global path planning to find the shortest path possible from the starting point to the end point and uses attitude estimation and plane plan to locate the visually impaired indoors. In addition, the system is equipped with human-computer interaction voice interface, which can not only receive audio instructions from the visually impaired, but also convert them into navigation information through voice recognition and send them to the server to start the path finding service. It can also guide the visually impaired by receiving navigation commands from the server. In order to improve the accuracy of visual SLAM in completing a certain task in navigation, S Agrawal et al. (2022) [20] proposed a visual SLAM smart blind cane with tactile navigation. The system is equipped with a depth camera and IMU for target positioning and path planning. The cane can identify empty chairs in public indoor places that conform to common social preferences and guide visually impaired to sit in these empty chairs. Tests show that the Visual SLAM cane was able to find a seat within 45 s on average, with an 83.3% success rate in finding an empty chair that matched social preferences. Although the Visual SLAM smart blind cane has the ability of visual navigation mapping, which shows good performance in the actual work, it still lacks the ability to obtain other information in the environment and cannot measure the distance of obstacles ahead.

#### 2.2.2. Visual Recognition Smart Cane

With the development of deep learning, many object recognition algorithms make machines learn rules from historical data to recognize objects through training and prediction. The smart cane with visual recognition function can effectively identify the obstacles in front of the visually impaired, or person’s face so that the visually impaired can get the information of the passers-by in front of him and identify the road conditions ahead (what type of road, whether there are holes in the ground and steps).

The visual recognition smart cane can identify different objects through different types of cameras. GE Legge et al. (2013) [45] developed a handheld sign reader based on infrared camera, which can identify two-dimensional bar codes of indoor data matrix and retrieve the location information, enabling the visually impaired to independently complete navigation tasks in indoor navigation. However, this kind of smart blind cane lacks the ability to identify obstacles, and is limited to the specific marked sites only, incapable of applying to most indoor and outdoor sites and is difficult to promote.

Most visual intelligence blinds canes need to recognize or detect one or more target objects to complete a specific task in a scene. In order to identify indoor obstacles, Cang Ye et al. (2014) [46] proposed smart blind cane using 3D-TOF camera for attitude estimation and object recognition. It can effectively detect indoor stairs, corridors and rooms and other architectural structures, as well as tables, computer monitors and other objects. At the same time, it can detect and estimate the action intention of the visually impaired, so as to switch among different working modes. Similarly, H Takizawa et al. (2015) [47] proposed a system composed of Kinect depth camera, digital keyboard and haptic feedback device. The system can identify objects such as chairs and stairs near the smart blind cane, search for the objects that the visually impaired wants to look for, and then feedback the search results to him through the haptic device. In order to identify outdoor vehicles and passers-by, A Carranza et al. (2022) [48] proposed a guide system composed of Raspberry Pi 4B official camera, ultrasonic sensor and speaker. The Raspberry Pi 4B official camera of the system uses the YOLO algorithm for real-time target detection, which can effectively identify pedestrians and vehicles in front of it. And the ultrasonic sensor is used to measure its distance to the object, and the information is fed back to the visually impaired by voice through the speaker. In order to detect obstacles and realize obstacle avoidance function, EJ Taylor (2017) [49] proposed an obstacle avoidance system for the visually impaired based on three-dimensional point cloud processing. Based on Robot Operating System (ROS) and open-source Point Cloud Library (PCL), the Asus Xtion Pro Live depth camera and wireless Wii remote control were mounted on the blind cane. Depth cameras can obtain semantic images of the visually impaired in front of him, to solve the obstacle avoidance problem. When the visually impaired is close to an obstacle, the blind cane system will warn the visually impaired, at the same time, it can warn the visually impaired there are obstacles above the waist. However, the smart cane is affected by the relatively narrow perspective of the vision sensor, and the visually impaired will encounter obstacles while walking, which poses a certain safety risk. In order to identify the track laid in advance on the ground and make the smart cane walk, TK Chuang et al. (2018) [50] proposed a guide dog type smart cane for the blind. Three cameras, three Raspberry Pi 2B and a Jetson TX1 were installed on the system. Deep networks from the real and virtual worlds function to overcome the visual system in the real-world environment of lighting, shadows, different background textures of challenging scenes. The three cameras are used for environmental data collection and only one camera input is used for prediction execution. The blind cane follows the pre-set track and leads the visually impaired to the destination. In order to recognize the face information in front of the visually impaired, Y Jin et al. (2015) [51] proposed a smart blind cane with the function of face detection and recognition. The system is equipped with a camera mounted on glasses and uses the Adaboost iterative algorithm to detect and recognize the faces of strangers around the visually impaired and generates the unique vibration pattern of each person to feed back to the visually impaired, helping him know who is in front of him.

#### 2.2.3. Obstacle Avoidance Smart Cane with Visual Integration with Other Sensors

It is difficult to complete the obstacle avoidance navigation task only by relying on a single visual sensor, because the visual sensor cannot obtain information such as the distance of obstacles in the environment around the visually impaired, his own motion posture and geographical position. Therefore, the visual smart cane has to be integrated with other sensor functions such as ultrasonic, infrared, IMU, GPS and other functions, and combines the recognition function of the visual sensor to enable the smart cane work with higher performance.

In order for the visual smart cane to be able to obtain more diverse information about the environment, researchers have made a number of attempts to integrate other sensors. The GPS function enables the visual smart cane to obtain location information. Adrien Brilhault et al. (2011) [28] proposed a smart cane that integrates GPS (global positioning system) and visual positioning, which can identify markers in the surrounding environment by using cameras and realize navigation function by using GPS. Similarly, MY Fan et al. (2014) [52] equipped an RGB-D depth camera, ultrasonic sensor, GPS and GPRS module on the blind cane. Using RGB-D depth camera to capture dynamic visual environment information, GPS and GPRS modules can obtain the target and the current position of the smart cane and upload the data to the remote center and using the ultrasonic module to help the visually impaired avoid obstacles. IMU enables the visual smart cane to obtain posture information. B Li et al. (2018) [53] combined the visual positioning service in Google Tango device and the pre-established voice map to realize the voice positioning of indoor environment. An obstacle detection and obstacle avoidance method based on TSM-KF algorithm and RGB-D camera is proposed, and a 9-axis IMU is used to track the relative direction of the smart blind cane and the blind person to achieve heading guidance. The visually impaired uses the keyboard to input the destination, and the Google Tango device will send the planned route to the smart cane through wireless Bluetooth technology. The smart cane can feed back voice and vibration information to guide the visually impaired to the destination. Similarly, H Zhang et al. (2019) [54] proposed to use the image and depth data of RGB-D depth camera and IMU inertia data to estimate the attitude of the smart blind cane for navigation path planning to reach the destination. At the same time, smart cane can also carry out human-computer interaction, with robot cane mode and ordinary mode, the visually impaired can switch among different modes according to the needs of the environment. The ultrasonic sensor can obtain the information of obstacles in the close range of the smart blind cane. MD Messaoudi et al. (2020) [55] proposed a smart blind cane integrating monocular camera, ultrasonic sensor, ESP32-WIFI module and voice module, as shown in Figure 4. The Pixy 2 monocular camera was mounted on the smart cane to detect colors, road signs and track lines. The ESP32-WIFI module mounted on the smart cane can transmit the data collected by the smart cane to the cloud through the Internet of Things technology, and receive data information such as path, obstacle position and traffic from the cloud.

#### 2.2.4. VR/MR Smart Cane

Virtual Reality (VR) and Mixed Reality (MR) systems can provide immersive visual images. The VR smart cane and MR smart cane are designed to immerse users in a virtual environment and perform realistic navigation tasks by providing visual and auditory feedback to the visually impaired.

J Kim et al. (2020) [30] proposed a VR smart blind cane, which is paired with VR glasses. By connecting the blind cane to the VR controller, the deep learning model enables the visually impaired to identify the blind path on the outdoor road and infer the direction of the blind path, providing a real walking experience for the visually impaired.

The MR Technology formed by adding (Augmented Reality) AR technology to VR technology has also been applied to the blind cane. D Tzovaras et al. (2009) [56] put forward a MR smart blind cane based on tactile and auditory feedback. The system is based on the use of cybergrabtm haptic equipment. An efficient collision detection algorithm based on hyperquadric surface is integrated to realize real-time collision detection in complex environments, and the visually impaired can navigate to the real-size virtual reality environment. L Zhang et al. (2020) [57] proposed a virtual blind cane that integrates VR and AR MR Technologies. The visually impaired person places an iPhone smartphone on a selfie cane and logs into a mobile VR app to access the virtual environment on the iPhone, simulating the blind cane in virtual reality. AR technology is applied to track the real-time posture of the iPhone smartphone in the real world, and then synchronize it with the blind cane in the virtual reality environment to identify the shape, size and location of the object in the virtual environment. Every time the virtual cane comes into contact with objects in VR, it will be converted into auditory and vibration and tactile feedback to the visually impaired.

However, the current VR/MR smart cane is still in the early stage of development, although it can realize more functions. However, the technology is still not mature, and more solutions need to be proposed to improve its practicability before it becomes a real guide device.

To sum up, the visual SLAM smart cane, the visual recognition smart cane, the visual fusion obstacle avoidance smart cane and the VR/MR smart cane all have their uniqueness and can play a role in the actual application, as shown in Table 2. However, in the face of more and more complex road conditions, it is necessary for the smart cane to have stronger adaptability to the environment with superior detection & navigation capabilities. The visual smart cane should integrate other sensors to improve the ability to obtain environmental information, efficiently identify obstacles ahead and have visual SLAM ability to meet the future road condition guide challenges.

## 3. Research Progress of Laser Vision Multi-Sensor Integration Smart Cane

In recent years, multi-sensor integration and multi-sensor fusion schemes have been introduced into the field of guide for the blind. Multi-source fusion schemes can obtain abundant ambient data and complete navigation and target recognition functions at the same time. The guide equipment attempts to integrate or fuse RGB-D depth camera with IMU inertial sensing unit technology [58], ultrasonic and vision technology [59], vision and infrared technology [60], millimeter wave radar and RGB-D depth camera technology [61], and RFID And GIS technology [62], etc., to improve the performance of the smart canes. Under the multi-source fusion scheme, the laser and vision fusion scheme has been increasingly applied to guide equipment (guide robot (guide suitcase) [36,63,64,65], virtual cane [66,67], wearable guide instrument [68,69]), greatly improving the overall performance of guide equipment. The laser sensor can play an important role in the detection and tracking of obstacles, but its sensitivity is not high in harsh environments (fog and rainy days) [70]. However, visual sensors such as depth cameras are usually used for semantic interpretation of scenes but cannot be used under the conditions of bad weather and illumination [71]. Therefore, laser vision can be complementary and integrated to balance the main shortcomings of each other. At the same time, the use of laser vision can reduce the local uncertainty of the guide system.

### 3.1. Laser Vision Integrated Sensor Smart Cane

The multi-sensor integration can reduce the uncertainty of the system to a certain extent and obtain multi-angle and multi-dimensional information with more accurate results according to different sensor observations. The schemes of multi-sensor integration and laser and vision integration have been applied in many guide equipment and proved their feasibility. The development of the smart cane is also influenced by the multi-sensor integration scheme and the laser and vision integration scheme. Researchers have integrated laser vision on the same smart cane.

The performance of laser sensing or vision sensing are limited, even though they are already providing services for the visually impaired, there are still many shortcomings. The laser and vision integrated sensing smart cane combines the performance of laser sensing smart cane and vision sensing smart cane, which can make up for the shortcomings of each of them to a certain extent. Fan K (2017) [72] et al. proposed a virtual blind cane combining a one-font linear laser emitter with a camera and adopted FPGA as a processing platform to realize obstacle ranging, detection and classification on embedded devices. However, this system can only measure the distance of a single point of the obstacle in front of the blind cane by using a linear laser emitter, and its detection range is too narrow to measure the rough outline of the obstacle. Moreover, the system lacks the ability to map the surrounding environment and cannot realize the active navigation function. Similarly, A Rahman et al. (2019) [73] proposed a smart blind cane scheme integrating a monocular camera and laser. The angle between the laser point and the monocular camera and the distance between the laser and the monocular camera are used to calculate the distance between the monocular camera and the obstacle. The system sends the feedback information to the visually impaired through voice and vibration, reminding the visually impaired to avoid obstacles or potholes on the road, at the same time, the system can also receive the location information of the visually impaired from the GPS.

Meanwhile, researchers also made relevant research on the sensor smart blind cane integrated with laser vision and other sensors, adding other environmental data sensors to the laser vision integrated sensor smart cane. QK Dang et al. (2016) [74] proposed a smart blind cane composed of camera, linear laser and Inertial Measurement Unit (IMU), as shown in Figure 5. The IMU is integrated into the smart blind cane composed of camera, linear laser and Inertial measurement Unit, and the posture and height information of the cane can be obtained. The system also carries out joint calibration of the camera and laser. When the laser line intersects the obstacle, the laser streak is generated, and the streak will be observed by the camera. The coordinates of each two-dimensional laser point observed in the camera image frame can be calculated relative to the camera coordinate frame in three-dimensional system position corresponding to the blind person, so as to estimate the three-dimensional position of the scanned object, and then track the pointing angle of the system to estimate the distance between the user and the obstacle. The cane uses voice and speech descriptions to feedback the detected distance of obstacles to the visually impaired. The sound feedback is a monophonic sound that is inversely proportional to the distance of the visually impaired to the object (the closer the distance, the louder the sound), while the sound description gives an indication of some special cases. Although the smart blind cane can calibrate and fuse the camera and the linear laser jointly, it has some problems such as low integration and is inconvenient to use. J Foucault et al. (2019) [75] proposed a portable environment-aware smart blind cane system named INSPEX, which is a blind cane system composed of Lidar, depth camera TOF sensor, ultrasound and ultra-wideband radar. The ultrasonic sensor and Uwiband radar system integrated with the smart cane can effectively improve the ability of ultra-short-range detection and environmental anti-interference. The system can effectively detect the obstacles around the blind cane, and its measuring range can be up to 10 m.

The smart blind cane with laser and vision integrated sensing combines the advantages of laser vision to effectively obtain more diverse environmental data around the visually impaired. The smart blind cane can be integrated with more sensors to have more functions. However, this kind of integrated blind cane is usually equipped with multiple sensors, so it has certain requirements in terms of integration. The higher the integration degree, the lighter the total system and the smaller the volume are bound to be the future development requirements of the laser and vision integrated sensing smart blind cane. However, we need to note that a higher integration degree does not mean a higher fusion degree. The system with high integration can effectively complement and optimize the combined processing of space information and produce consistent analysis of the observed environment. Therefore, we can improve the coupling of sensors and the fusion of algorithms to enhance the robustness of the smart blind cane system, and the smart blind cane with higher integration will certainly have better sensing performance.

### 3.2. Comparative Discussion

The comparison of the performance of laser sensing smart cane, vision sensing intelligent blind cane, laser and vision integrated sensing smart cane, laser and vision fusion sensing smart cane is shown in Table 3. The laser sensing smart cane can measure the distance of obstacles ahead but cannot identify them, while the visual sensing smart cane can identify obstacles ahead but cannot measure the distance of obstacles. Therefore, the smart cane that only relies on a single sensor lacks some functions. The laser and vision integrated sensing smart cane can effectively integrate the functions of laser sensing and vision sensing, but the loose coupling between the sensors leads to the low efficiency of the system. Laser and vision fusion sensor smart cane can simultaneously have ranged and recognition functions, and the tight coupling between the sensors can improve the efficiency of the system. Therefore, the laser and vision fusion sensor smart cane have a variety of superior properties, so it is worth to carry out research and development in this direction to bring more superior performance of the smart cane for the visually impaired.

### 3.3. Smart Canes Applying the Laser and Vision Fusion Sensing Technology

Lidar can obtain ambient point cloud data, while visual cameras can obtain ambient images. Based on laser and vision sensors, sensor data can be used in the automatic system to complete tasks such as automatic driving, target detection, simultaneous positioning and composition (SLAM) [76], mapping, target tracking, real 3D reconstruction, etc. The poor robustness of visual perception system under complex lighting conditions severely limits its all-weather working ability. However, Lidar based systems are not affected by lighting conditions and can quickly and efficiently provide high-resolution 3D geometric information of the environment. Nevertheless, due to their low resolution, low refresh rate and high cost of its high-resolution sensors, Lidar sensors are also greatly affected by target velocity drift, different target sizes, local occlusion and so on. Since Lidar can provide accurate three-dimensional point cloud geometry, while cameras capture more scene images and semantic information, the fusion of two different sensors is now a mainstream technology development for better performance. This chapter briefly describes the research progress of laser and vision fusion technology, and finally discusses the challenges of laser and vision fusion technology research and application as well as the future development potential, and the application of laser and vision fusion sensor technology onto the smart canes, with superior navigation and target recognition capability.

#### 3.3.1. Laser and Visual Fusion SLAM

At present, the mainstream navigation schemes of various mobile platform applications are visual SLAM and laser SLAM. In the process of development, navigation schemes of multi-source fusion SLAM and laser and visual fusion SLAM have also been proposed accordingly. Laser SLAM has always been the dominant navigation technology of mobile platforms, and the reliability and security of laser SLAM is better than that of visual SLAM. Laser SLAM can realize stronger anti-interference performance by relying on the good penetration and monochromism of laser detection light source. Therefore, laser SLAM has the advantages of high precision, fast speed and rapid response to environmental changes under dynamic and static conditions. However, laser SLAM is limited by the detection range of Lidar, with poor relocation ability, not suitable for working in similar geometric environments such as long straight corridors and having weak adaptability to scenes with large dynamic changes. While visual SLAM is a mainstream technology in the future, which is better than laser SLAM in terms of cost. Visual SLAM has the advantages of no limitation of sensing detection range and can extract the semantic information of the map. But the visual camera is greatly affected by the lighting conditions, accurate environment image information can not be obtained under too strong light or dim environments. If only based on visual SLAM, the accuracy of map construction will be low, and there will be cumulative errors in map construction. The laser radar has sparse and high-precision depth data, and the camera has dense but low-precision depth data. The fusion of the two can complete the restoration of pixel depth in the image. Ji Zhang et al. (2015) [77] proposed a V-Loam system combining visual odometer and Lidar data. Based on visual odometer and scanning matching Lidar odometer, the performance of real-time motion estimation and point cloud registration algorithm is improved. Shao et al. (2019) proposed VIL-SLAM [78] that combines tightly coupled stereo VIO with Lidar mapping and Lidar enhanced visual loop closure. X Zuo et al. (2019) [79] proposed the LIC-Fusion that integrates IMU data, sparse visual features and extracted Lidar point cloud data.

Laser and visual fusion SLAM are definitely a better localization and navigation solution. Therefore, through the fusion of laser and visual SLAM, the advantages of laser SLAM and visual SLAM are combined to make up for their shortcomings. The development of smart cane equipped with laser and vision fusion SLAM sensing system can help improve the ability of autonomous positioning, obstacle avoidance and navigation of the smart cane.

#### 3.3.2. Laser and Vision Fusion Target Detection

Target detection methods based on Lidar include feature extraction, raster map and so on. The feature extraction method is mainly to cluster the obstacle point cloud data, then extract the feature information of the obstacle from the point cloud within the cluster, and finally analyze the feature information to realize the detection and recognition of the obstacles ahead. The raster map method is mainly to raster point cloud data and generate local raster map, and then conduct raster clustering, and cluster the grids in adjacent areas into a class to complete the object detection of obstacles. Through image detection based on vision sensor, the position and category of image target can be obtained. Target detection methods based on vision sensors include traditional target detection algorithms and deep learning target detection algorithms. Traditional target detection algorithm uses Windows of different sizes to slide, to obtain region of interest (ROI), and then to carry out feature extraction on these regions. Finally, classifiers are used to classify the extracted features to detect target obstacles. Since it is difficult for single laser vision to realize the perception of the surrounding environment and the object detection of obstacles, laser sensors and vision sensors have good complementarity. Laser and vision fusion target detection method can effectively improve the type and accuracy of target detection.

In laser and vision fusion target detection, YOLO algorithm is widely used because of its fast-running speed and can be used in real-time system. Rovid et al. (2020) [80] proposed a neural network-based approach to fuse visual images and Lidar point clouds. Each 3D point in the lidar point cloud is enhanced by semantic strong image features, and the input is connected to the F-PointNet neural network architecture. More accurate detection of cars, pedestrians and cyclists can be realized using the Tiny Yolo3 model, as shown in Figure 6. Mingchi F et al. (2021) [81] used YOLOv3 and point-GNN networks to detect image and form Point cloud, respectively. Secondly, in order to determine the shape and position of the object, a method of shape and position estimation using the 3D point and type of the object is proposed. The image object and point cloud object are transformed to the same vehicle coordinate system by using the calibrated external reference relation. The global nearest neighbor (GNN) data association method was used to match both image objects and point cloud objects, and the extended Kalman filter (EKF) algorithm was used to track the matched objects.

There are limitations in target detection by using vision sensor or Lidar alone. Therefore, the development of smart cane equipped with laser and vision fusion target detection system can help improve the ability of smart cane to identify obstacles and meet the real-time requirements of smart cane in navigation and recognition.

#### 3.3.3. Laser and Vision Fusion Sensor Smart Cane

The above research shows that the fusion of Lidar and visual camera can effectively improve the accuracy of navigation map construction and target recognition. Sensor fusion can be carried out in three levels: data preprocessing, feature extraction and decision stage, but the data fusion is easily limited by the information conflict between different sensors, and the decision stage fusion technology is simple and unreliable. Therefore, we believe that the dynamic noise in subgraphs can be eliminated by map generation and fusion based on d-s evidence theory fusion in decision stage, and the reliability of YOLOV3 boundary frame association in target recognition can be increased based on d-s evidence theory.

Therefore, we put forward a novel laser and vision fusion sensor smart blind cane that is of practical value and made efforts to develop the actual device. Laser and vision fusion sensor smart blind cane is equipped with laser and vision fusion SLAM system and laser and vision fusion target detection system. In the aspect of laser and vision fusion SLAM, the dynamic noise in subgraphs is eliminated by map generation and fusion based on d-s evidence theory fusion at the decision stage, and Lidar and RGB-D cameras are designed by tight coupling structure and field of view structure adapted to the environment. Fusion map construction was carried out with Cartographer/ORB-SLAM2 algorithm, and spatial and temporal synchronous laser and visual SLAM sensing system was developed. In terms of laser and vision fusion target detection, d-s evidence theory is used to increase the reliability of YOLOV3 boundary frame association in target recognition. The target point cloud cluster on the front object target is extracted by Lidar point cloud, and the target point cloud cluster is projected into the spatially aligned visual image. The corresponding ROI was obtained, and then the optimized YOLOV3 target detection network was used for target detection and classification recognition. The developed laser and vision fusion system and target detection system are mounted on the intelligent blind cane at the same time, so that the smart blind cane can obtain active navigation and target recognition functions. The Laser and vision fusion sensor smart cane scheme is shown in Figure 7. In order to reduce the overall weight of the smart cane, the low-cost and small size of two-dimensional Lidar, ORBBEC Gemini depth camera, NVIDIA jetson nano, STM32 micro-controller were adopted, to enhance the coupling between sensors and increase the flexibility of the smart cane movement.

## 4. Future Development of Laser Vision Sensing Smart Canes

With the government’s policy help for the visually impaired and other vulnerable groups, guide equipment technology is developing rapidly in the direction of intelligence and information capabilities, and the visually impaired people have a stronger desire to improve their independence, travel conveniently and safely, and integrate into social life. Therefore, higher requirements are put forward for experimental research and development. The smart cane with integrated multi-sensor guide system which can adapt to complex terrain environment and bad weather, with active navigation leading ability and coordination ability of the Internet of Things (IoT) will inevitably become the development trend.

### 4.1. Integration and Fusion of Multi-Sensor Smart Canes

Nowadays, smart cane sensor tends to develop in the direction of multi-sensor integration and fusion. Although the laser vision has superior performance, they still need to face many difficulties and challenges for the visually impaired to travel freely. Therefore, the laser and vision cane can also add IMU (inertial sensing unit), GPS, GIS, solar charging functions to enhance the navigation performance and endurance of the laser and vision smart cane. Therefore, in the future, laser vision sensing intelligence will also develop towards multi-source fusion of smart canes, with improved coupling between smart cane sensors and reduction in the mutual interference among sensors. At the same time, the laser and visual sensing smart cane will be developing towards the direction of high degree of integration, reducing the overall weight, and increasing its mobility.

Highly integrated embedded smart cane is easy to dismount, which is conducive to the visually impaired in some special road conditions (such as pedestrians and roads with heavy traffic, bumpy ground, stairs with a number of steps), and in some bad weather not suitable for blind cane navigation (foggy and rainy days, snow days, night conditions). The guide device system on the smart cane can be removed and put in the backpack to make the smart cane become a common cane to cope with special road conditions and bad weather, to reduce the weight of the smart cane and increase the flexibility of the smart cane to ensure the personal safety of the visually impaired.

### 4.2. Navigation Modes of Smart Canes

There are two kinds of navigation modes: passive navigation and active navigation. The smart cane of passive navigation will feed back the environmental information around to the visually impaired through the feedback module and inform the visually impaired how to walk by means of touch, vibration, voice and sound. Therefore, passive navigation requires the visually impaired to understand the road conditions according to the feedback information before walking. At this time, the passive navigation smart blind cane acts as an environmental sensing assistant and cannot guide the visually impaired to the destination actively like the “guide dog”. The smart cane for active navigation is, however, like a guide robot with a shaped cane. The cane is equipped with a roller, omnidirectional wheel or a device similar to a robot car. The active navigation smart blind cane can not only feed back environmental information to the visually impaired, but also navigate him to the destination with the help of wheels, thus reducing the time for him to follow instructions and analyze and judge and improving the travel efficiency. The navigation modes of the smart blind cane are shown in Table 4.

With the rapid development of automatic driving and robot navigation technology, the smart cane will develop towards the active navigation to improve the travel speed of the visually impaired, in which the smart cane plays the role of “guide dog” with the navigation technology similar to automatic driving and robots.

### 4.3. Smart Cane System Coordinated by IoT

Some researchers have carried out research on the smart cane based on the Internet of Things technology (IoT) [55,83]. The information interaction between the smart cane and the visually impaired, pedestrians and vehicles on the road, nearby buildings and infrastructure will weave the guide system of the whole society into a network of mutual connections. At this time, each smart cane is both the information sharer and the information recipient, and it can voluntarily share the nearby environmental data acquired to other nearby smart canes, and also accept the environmental data shared by other smart canes or vehicles, so as to form a coordinated, stable and effective smart guide system ecology of IoT. Using this strategy, it is more beneficial to improve the travel efficiency of the visually impaired and ensure his or her safety.

## 5. Conclusions and Discussion

At present, most of the laser vision sensing smart blind canes are still in the stage of theoretical scheme design and prototype research and development. Many core technologies still need to be developed fully, and there are still many auxiliary functions, software and hardware design and shape structure, psychological problems of the visually impaired [84] and travel needs of the visually impaired [85] to be further studied and explored. However, we believe that with the joint efforts of the global smart cane researchers, people from all walks of life are more supportive to the visually impaired and other groups. The smart canes will develop rapidly towards an integrated, portable, low-cost, humanized and multi-application scenarios direction. Laser and vision sensing technology is still developing. In the future, laser and vision sensing technology can play an even more important role in the smart canes, enabling them to obtain more detailed environmental data with better navigation ability. Looking ahead, we will continue to be committed to the development of laser and vision fusion SLAM and target recognition system that can be mounted on the smart cane, so that the smart cane can integrate laser sensing technology and visual sensing technology, enabling it to have the function of active navigation and guidance, efficient recognition and detection of targets. In this way, not only the walking speed of the visually impaired can be enhanced, but the types and positions of obstacles in front of them can be effectively understood, so as to provide safe and efficient travel guarantee for the visually impaired. This, along with other endeavors in caring for the disadvantaged groups, will certainly contribute to the building of our harmonious society.

## Figures and Tables

**Figure 1 sensors-23-00869-f001:**
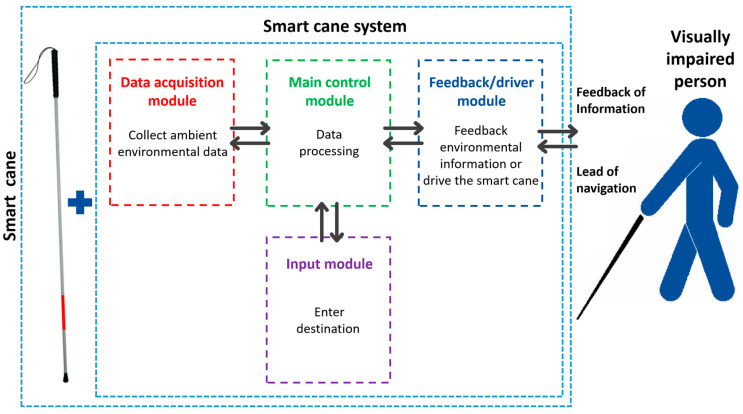
Structure diagram of smart cane.

**Figure 2 sensors-23-00869-f002:**
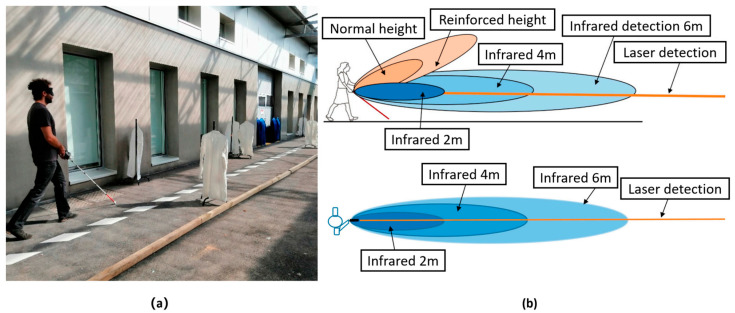
(**a**) A visually impaired holding the Tom Pouce III laser smart cane; (**b**) the detection range of the smart cane [37].

**Figure 3 sensors-23-00869-f003:**
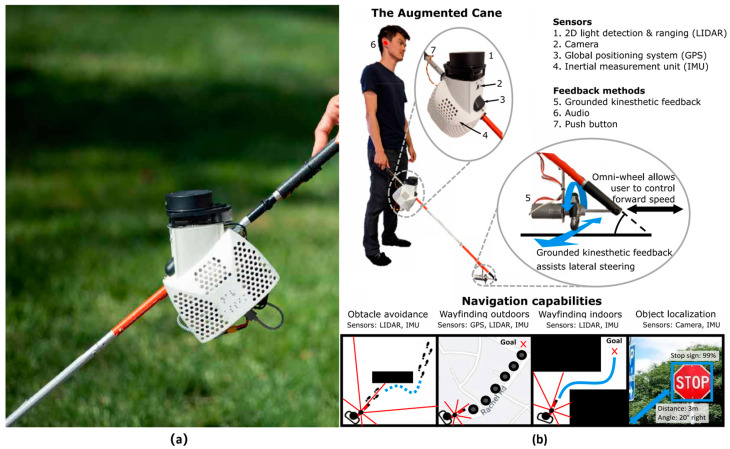
(**a**) Laser SLAM smart cane; (**b**) System diagram of the smart cane [21].

**Figure 4 sensors-23-00869-f004:**
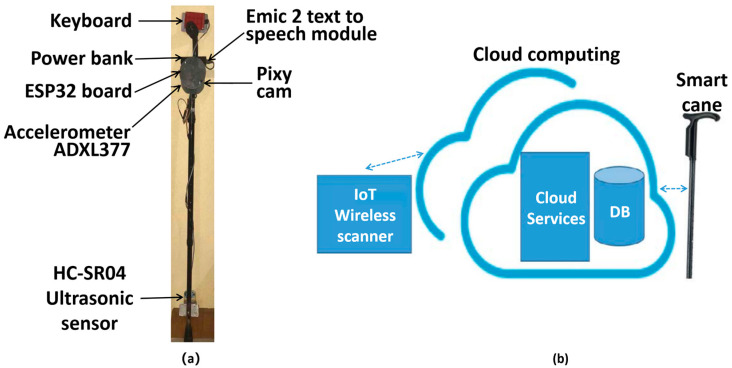
(**a**) The components of the Smart Cane; (**b**) The system structure of smart cane [55].

**Figure 5 sensors-23-00869-f005:**
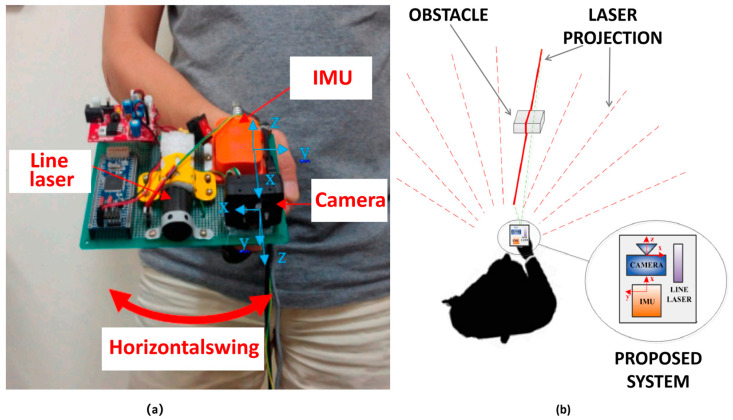
(**a**) Smart cane composed of camera, linear laser and IMU; (**b**) Working diagram of the smart cane system [74].

**Figure 6 sensors-23-00869-f006:**
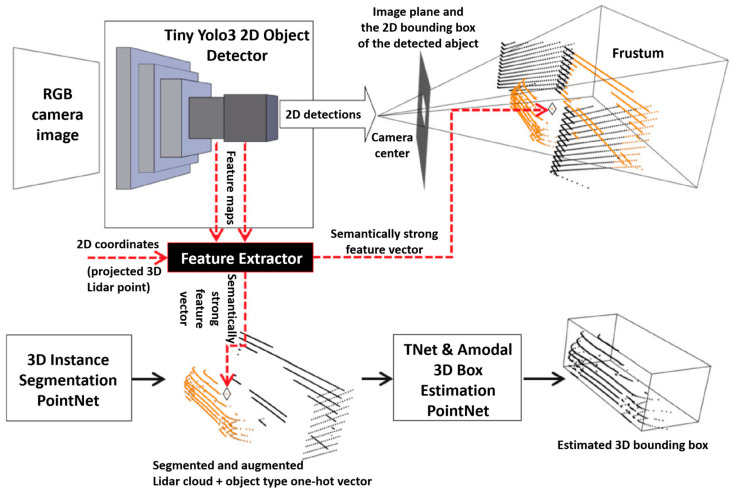
Target detection architecture of Lidar and camera fusion [80].

**Figure 7 sensors-23-00869-f007:**
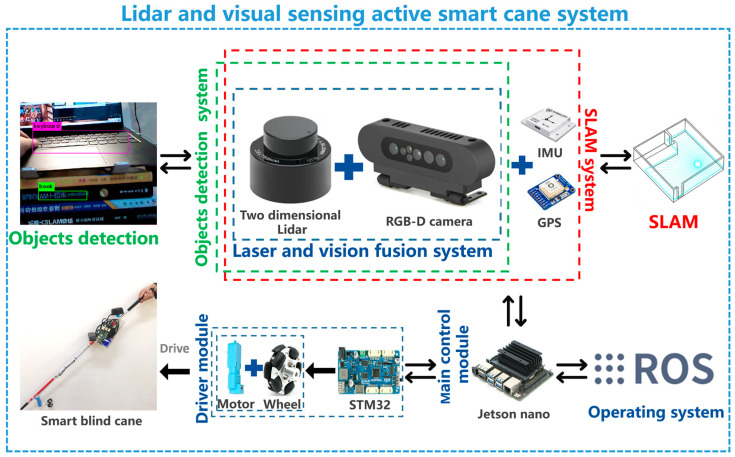
Lidar and visual sensing active smart cane system.

**Table 1 sensors-23-00869-t001:** A short summary of laser sensing smart canes.

Type	Sensor	Advantages	Disadvantages	References
Laser ranging	Laser rangefinder, laser scanner	High ranging accuracy	Unable to measure surrounding obstacle distance	[22]
Lidar ranging	Lidar	Measure the distance of obstacles around	High price, low endurance	[37,38]
Virtual	laser scanner	Light weight, flexible to use	Cannot be used in difficult road conditions	[31]
Integrated sensor	Laser Scanner + IMU	Obtain attitude information	(1) Loose coupling (2) Large volume (3) Relatively heavy	[39]
	Laser rangefinder + ultrasonic	Increase range		[40]
Laser SLAM	Lidar	Map navigation	Cumulative map error	[21]

**Table 2 sensors-23-00869-t002:** Summary of various Visual sensing smart canes.

Type	Advantages	Disadvantages	References
Visual SLAM	Path planning	(1) Need prior maps (2) No obstacle recognition function (3) Unable to measure distance	[20,41,42,43,44]
Visual recognition	Identify obstacles	(1) Unable path planning (2) Unable to measure distance	[45,46,47,48,49,50,51]
Integrated sensor	Get more performance	Loose coupling	[28,52,53,54,55]
VR/MR	Perceive the outside world	(1) Immature technology (2) Unable to measure distance	[30,56,57]

**Table 3 sensors-23-00869-t003:** Laser sensing, vision sensing, integrated sensing, fusion sensing smart canes.

Type	Advantages	Disadvantages	References
Laser sensing	Measuring distance	Non-recognition function	[37]
Visual sensing	Recognition function	No distance measurement function	[47]
Integrated laser and visual sensing	Measuring distance and recognition function	Sensor loose coupling	[75]
Laser and visual fusion sensing	(1) Measuring distance and recognition function (2) Sensor tight coupling	Immature technology	[74]

**Table 4 sensors-23-00869-t004:** Navigation modes of smart canes.

Mode of Navigation	Advantages	Disadvantages	References
Active navigation	(1) Lead of navigation (2) Fast response speed	(1) High performance requirements for CPU and sensor (2) High power consumption	Omnidirectional wheel [21], Robot car [82]
Passive navigation	Feedback environmental information	No navigational lead	Tactile [43], vibration [36], speech [44], sound [40]

## Data Availability

Not applicable.

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
