# Peer review of "Laser Sensing and Vision Sensing Smart Blind Cane: A Review"

_sensors, 2023, doi:10.3390/s23020869_

Round 1

Reviewer 1 Report

Update the references and introduction 

Reviewer 2 Report

Comments:

1. The novelty of this study should be inserted in the text clearly. 2. The advantages and disadvantages of this study should be investigated.    

Reviewer 3 Report

The authors have done an extensive review on blind cane.

Very good work, useful to the society.

Language is good.

By adding cameras and other sensors the cost of the equipment increases, is it affordable to the common man and at the same time the blind person has to take care of the equipment from thieves. What are your solutions for these?

Reviewer 4 Report

Greetings, Editor thank you for providing me with the opportunity to review the article. I reviewed the article with title = Review of laser vision sensing smart blind cane. Overall, the article structure and content are suitable for the sensors  journal. I am pleased to send you major level comments. Please consider these suggestions as listed below.  

  1. The title seems weird. Please change
  2. The abstract seems to be good. Please add one more introductory line of your objective in beginning of abstract.
  3. Research gap should be delivered on more clear way with directed necessity for the future research work.
  4. Introduction section must be written on more quality way, i.e., more up-to-date references addressed.
  5. The novelty of the work must be clearly addressed and discussed, compare previous research with existing research findings and highlight novelty.
  6. What is the main challenge? Why there is need of this review?
  7. Please check the abbreviations of words throughout the article. All should be consistent.
  8. What is problem statement?
  9. The main objective of the work must be written on the more clear and more concise way at the end of introduction section.
  10. Please provide space between number and units. Please revise your paper accordingly since some issue occurs on several spots in the paper.
  11. Overall discussion sections are well explained.
  12. Please add a comparative discussion section. It would be better for reader.
  13. Conclusion and Future perspectives should be added in section 6. Conclusion section is missing some perspective related to the future research work, quantify main research findings, highlight relevance of the work with respect to the field aspect.
  14. To avoid grammar and linguistic mistakes, major level English language should be thoroughly checked. Please revise your paper accordingly since several language issue occurs on several spots in the paper.
  15. Reference formatting need carefully revision. All must be consistent in one formate. Please follow the journal guidelines.

Round 2

Reviewer 4 Report

The revised version is well improved. I am in favour to accept in the present form.

Author Response

Thank you for your suggestions!